# Knowledge, attitudes, and practices toward the novel coronavirus among Bangladeshis: Implications for mitigation measures

**Alak Paul**[1], **Dwaipayan Sikdar**[2], **Mohammad Mosharraf Hossain**[3], **Md Robed Amin**[4], **Farah Deeba**[5], **Janardan Mahanta**[6], **Md. Akib Jabed**[1], **Mohammad Mohaiminul Islam**[1], **Sharifa Jahan Noon**[1], **Tapan Kumar Nath**[7] *

1 Department of Geography and Environmental Studies, University of Chittagong, Chittagong, Bangladesh, 2 Department of Biochemistry and Molecular Biology, University of Chittagong, Chittagong, Bangladesh, 3 Institute of Forestry and Environmental Sciences, University of Chittagong, Chittagong, Bangladesh, 4 Department of Medicine, Dhaka Medical College, Dhaka, Bangladesh, 5 Honorary Post-Doctoral Fellow at Macquarie University (Australia) and Department of Clinical Psychology, University of Dhaka, Dhaka, Bangladesh, 6 Department of Statistics, University of Chittagong, Chittagong, Bangladesh, 7 School of Environmental and Geographical Sciences, University of Nottingham Malaysia, Semenyih, Selangor, Malaysia

* Tapan.Nath@nottingham.edu.my

**Data Availability Statement:** Raw data of this work has been uploaded in this submission as supplementary information

## Abstract

The current novel coronavirus (nCoV) pandemic, COVID-19, was first reported in December 2019 in Wuhan, China, and has spread globally, causing startling loss of life, stalling the global economy, and disrupting social life. One of the challenges to contain COVID-19 is convincing people to adopt personal hygiene, social distancing, and self-quarantine practices that are related to knowledge, attitudes, and practices (KAP) of the residents of respective countries. Bangladesh, a densely populated country with a fast-growing economy and moderate literacy rate, has shown many hiccups in its efforts to implement COVID-19 policies. Understanding KAP may help policy makers produce informed decisions. This study assessed KAP in relation to COVID-19 in Bangladesh. An online survey using a pre-tested questionnaire conducted in late March 2020 attained 1,837 responses across Bangladesh. Ultimately, 1,589 completed responses were included in a statistical analysis to calculate KAP scores and their interrelations with sociodemographic variables. The overall KAP was poor, with only 33% of the participants demonstrating good knowledge, whereas 52.4% and 44.8% of the subjects showed good attitudes and practices, respectively. Sociodemographic factors had strong bearings on the KAP scores. Significantly higher KAP scores were evident in females over males, among aged 45 years and older over younger participants, and among retired workers and homemakers over students and public service employees. This study indicated a panic fuelled by poor understanding of COVID-19 associated facts and the need for the government to ensure more granular and targeted awareness campaigns in a transparent and factual manner to foster public confidence and ensure more meaningful public participation in mitigation measures. This study provides a KAP baseline regarding COVID-19 among Bangladeshis.

**Funding:** The author(s) recieved no specific funding for this work.

**Competing interests:** The authors have declared that no competing interests exist.

## Introduction

The rapidly unfolding coronavirus disease 2019 (COVID-19) pandemic has disrupted life globally. The novel coronavirus (nCOV, later called SARS-CoV-2) originated from an unknown source in Wuhan, China [1–3]. Unlike previous coronavirus outbreaks [4], this highly contagious [5–9] zoonotic virus from an as-yet-unconfirmed animal origin [10,11] evolved from a local flu-related severe acute respiratory syndrome [4,8,12,13] to a pandemic threatening the lives of millions within a few weeks. COVID-19 has thrown global public health into turmoil by severely straining many nations' healthcare systems. The epicenter rapidly moved from China to Iran and then through Europe and the US over a span of nine weeks [14]. As it spread through social contact [15,16], billions were forced into lockdown to minimize the transmission rate [4]. Lockdowns were necessary since researchers need time to develop a vaccine or effective treatment as in preceding pandemics including SARS and MARS [4,17,18]. No imminent solution for COVID-19 is likely in the immediate future [19].

The first-world healthcare system has failed to provide medical care for the rapidly increasing number of infected patients, let alone developing or underdeveloped nations [20,21]. In the majority of the cases, the leadership and bureaucracy in different countries seemed indecisive, inefficient, unprepared, and unable to contain the contagion. For the first time in history, the active participation of every single person on earth, in the form of testing, isolation, contact tracing, social distancing, staying at home, self-quarantining, improving personal hygiene, and using personal protective equipment such as masks and gloves, has become critical to contain COVID-19, prevent healthcare workers from becoming overwhelmed, and give researchers time to develop treatment strategies [22,23]. Hundreds of millions have sacrificed their autonomy, health, job, business, recreation, and education. However, ensuring voluntary participation in COVID-19 prevention strategies has posed challenges in different countries due to varying levels of knowledge, attitudes, and practices (KAP). Accordingly, the design and success of anti-contagion initiatives depend on macro- and micro-level understanding of KAP in respective regions and within each country.

In Bangladesh, similar to many nations in the SAARC region, COVID-19 seems grave [24,25] mainly due to cases imported by expatriates [26]. Following its first positive COVID-19 case on March 8, 2020 [27], Bangladesh shuttered its educational institutions on March 17, saw its first COVID-19 death on March 18 [28], and instituted a nationwide lockdown on March 26. Law enforcement and the army were mobilized to strengthen the lockdown's implementation as Bangladesh is densely populated and depends on labor-oriented industries [29], and a vast majority of its residents subsist on daily earnings through informal occupations [30]. However, the lack of a coordinated response to the threat of COVID-19 is evident [27], indicating that the design and implementation of these initiatives was based on a poor understanding of various sociodemographic groups' KAP. Hence, this study assessed the knowledge, attitudes, and practices of Bangladeshis on nCOV using netizens as a representative sample. We hope that the outcomes will assist authorities and other stakeholders to improve the planning and execution of different measures and provide a reference for countries with similar sociodemographic characteristics.

## Methods

### Conceptual framework

This study followed the KAP approach because this is a representative tool used for specific populations to collect information on what is known, believed, and done in relation to a specific field, for example, health [31]. Historically, the KAP model was developed for family planning and population studies in the 1950s. KAP was used to measure the extent to which any

clear opposition to family planning existed among different populations, so specific family planning practices could be used for different programs worldwide [31]. KAP surveys are now the most widely used studies for demonstrating societal context in public health research [32–34]. These surveys are easy to design, data output is quantifiable, interpretation is robust, and their utility is generalizable for context-specific problems [35]. The information generated through KAP studies can be used to develop strategies with a focus on improving the behavioral and attitudinal changes driven by the level of knowledge and perceptions toward preventive practices [36]. In a recent KAP study conducted in China, Zhong et al. [37] reported that to facilitate outbreak management, it is urgently necessary to understand the public's awareness of COVID-19. They asserted that success against COVID-19 requires peoples' adherence to control measures that is largely affected by their knowledge, attitudes, and practices.

## Instruments and participants

This KAP study was conducted across Bangladesh using an online survey. Because of the contagious nature of COVID-19, we avoided physical interviews. Following Zhong et al. [37], who studied KAP in COVID-19 infected areas in China, we prepared a structured questionnaire with 50 multiple-choice questions. It was tested in a pilot study with 10 participants. Based on feedback from the pilot study, we revised the questionnaire and finalized it with 40 questions (S1 Table). The questionnaire had four parts: A) basic participants' information (5 questions), B) COVID-19 knowledge (16 questions), C) attitudes (10 questions), and D) practices (9 questions). Using Microsoft Office 365, a form was created and a link to the form was shared through Facebook and email with a brief introduction and the survey objectives. Prospective participants were asked to share the form widely to collect a snowball sample of representatives of Bangladeshi netizens aged 18 and older. The survey anticipated responses from participants with university-level education as the questions were written in English, and the participants were the authors' Facebook friends and friends of friends who were mostly university graduates and students. As such, this study's population was unknown and therefore it was not possible to estimate the response rate and sample size before data collection. Participation was voluntary and anonymous, and the subjects could withdraw from the survey at any time. Before participating in the survey, prospective participants had to answer a yes/no question to confirm their consent to participate voluntarily. By answering the yes question, the participants provided informed consent prior to complete the survey. After providing their consent, the participants were directed to complete the questionnaire. The form was posted on March 22, 2020, at 22:00, and the survey was closed on March 28, 2020, at 00:15.

To contextualize the participants' views with the public sectors' preparedness for COVID-19, mitigation-relevant policy documents, press releases, and newspaper reports were reviewed, synthesized, and described following content analysis [38]. The ethical review committee of Dhaka Medical College, Bangladesh, approved this survey (memo no. ERC-DMC/ECC/2020/88).

## Data cleaning

The participants input their opinions and information using the shared online survey form. Their responses were automatically stored in Microsoft Excel. A total of 1,837 subjects participated in this study. However, some did not fully complete the survey questionnaire. Incomplete responses were discarded, leaving 1,589 complete responses.

## Scoring method

Respective knowledge, attitudes, and practices' scores for each respondent were obtained from their responses respectively on 13 knowledge questions, 10 attitude questions, and 8 practice

questions. The percentage of correct answers on knowledge, attitudes, and practices questions yielded the scores of the respective categories. A cut point of 80% correct answers was used for all of the categories to differentiate between *good* and *poor* knowledge, attitudes, and practices (S2 Table).

## Software

The online survey was conducted by distributing the KAP questionnaire as a Microsoft Office form through Facebook and email. After importing the online survey results through Microsoft Excel, R version 3.5.2 was used for raw data management and statistical analysis. Some statistical analyses were conducted using SPSS (Statistical Package for the Social Sciences) version 16.

## Statistical analysis

Scores of questions on knowledge (13 questions), attitudes (10 questions), and practices (8 questions) were estimated using a score of 1 for each right answer, 0.5 for each maybe answer, and 0 for each incorrect answer. The percentage of correct/maybe answers on knowledge, attitudes, and practices were those categories' scores. Based on the different knowledge, attitudes, and practices' variables scores, the mean difference between/among the categories of different sociodemographic characteristics was compared using the independent sample t-test (for two categories of variables) and one-way analysis of variance (ANOVA)/F-test (for more than two categories of variables). The associations between different knowledge, attitudes, and practices' variables with different sociodemographic variables were shown using the chi-squared test. Logistic regressions were run on the significant variables in the bivariate analyses/chi-squared test. Some attitude and practice variables with more than two categories were grouped into two categories, "yes" for all "yes" responses and "others" for all "no" and "maybe" responses to several questions (S3 Table). The "yes" and "others" categories were used for regression analysis of these variables.

## Results

### Sociodemographic characteristics of the participants

The study participants' characteristics are summarized in Table 1. The majority were males (60.48%) and 18–25 years old (46.5%), indicating that the online survey disproportionately reached a younger population. As expected, most of the participants (95.78%) had a university-level education. By occupation, 44.5% of the participants were students, followed by professionals (40.3%).

### Bangladeshi netizens' COVID-19 KAP scores

As we considered a cut point of 80%, the participants' overall knowledge score was poor, with a mean score of 9.60±1.45 on a scale of 13.0 (Table 2). There was no statistically significant difference in knowledge scores between males (9.65±1.49) and females (9.52±1.38), a good outcome due to the wide range of educational support for females in Bangladesh. Surprisingly, the difference in knowledge scores was also insignificant between educational groups. Conversely, age and occupation had statistically significant (p<0.01) effect on knowledge scores. Among the age groups, older participants were more knowledgeable on COVID-19 than younger participants. Retirees had significantly higher knowledge scores (10.55±1.37) than the other occupation groups. Contrary to expectations, students had poor knowledge scores (9.35±1.45), second only to homemakers, who had the lowest scores (9.29±1.39).

**Table 1. Sociodemographic profile of the participants.**

| Categories | Groups | Frequency | Percentage |
|---|---|---|---|
| Gender | | | |
| | Male | 961 | 60.5 |
| | Female | 628 | 39.5 |
| Age (years) | | | |
| | 18–25 | 739 | 46.5 |
| | 26–35 | 533 | 33.5 |
| | 36–45 | 184 | 11.6 |
| | Over 45 | 133 | 8.4 |
| Education | | | |
| | Secondary & Below | 67 | 4.2 |
| | University | 1522 | 95.8 |
| Occupation | | | |
| | Government staff | 73 | 4.6 |
| | Homemakers | 43 | 2.7 |
| | Professionals | 640 | 40.3 |
| | Retired | 11 | 0.7 |
| | Student | 707 | 44.5 |
| | Unemployed | 115 | 7.2 |

The mean attitude scores of 8.16±1.07 on a scale of 10 indicated that the participants had the desired attitude toward COVID-19 (Table 2). This score varied significantly between genders ($p<0.01$) and among occupation groups ($p<0.05$). The females' attitude score (8.34±1.00) was higher than the males' score (8.04±1.11), although both had comparable knowledge scores.

**Table 2. Sociodemographic distribution of the participants and their KAP scores.**

| Demographic variables | | Knowledge score in 13.00 ($\bar{x} \pm s$) | t/F test | Attitude score in 9.00 ($\bar{x} \pm s$) | t/F test | Practice score in 9.00 ($\bar{x} \pm s$) | t/F test |
|---|---|---|---|---|---|---|---|
| Gender | | | | | | | |
| | Male | 9.65±1.49 | 1.79 | 8.04±1.11 | -5.48*** | 6.03±1.27 | -6.43*** |
| | Female | 9.52±1.38 | | 8.34±1.00 | | 6.44±1.16 | |
| Age (years) | | | | | | | |
| | 18–25 | 9.34±1.47 | 18.66*** | 8.13±1.11 | 1.33 | 6.21±1.25 | 0.927 |
| | 26–35 | 9.76±1.37 | | 8.23±1.04 | | 6.21±1.22 | |
| | 36–45 | 9.77±1.47 | | 8.10±1.11 | | 6.18±1.29 | |
| | Over 45 | 10.17±1.33 | | 8.12±0.97 | | 6.03±1.18 | |
| Education | | | | | | | |
| | ≤12 years | 9.38±1.41 | -1.26 | 8.01±1.12 | -1.13 | 5.93±1.20 | -1.76 |
| | >12 years | 9.61±1.45 | | 8.17±1.07 | | 6.21±1.24 | |
| Occupation | | | | | | | |
| | Govt. Staff | 9.72±1.42 | 10.37*** | 7.94±1.31 | 3.049*** | 5.70±1.23 | 4.12*** |
| | Homemakers | 9.29±1.39 | | 8.42±0.76 | | 6.55±1.12 | |
| | Professional | 9.87±1.36 | | 8.25±0.97 | | 6.16±1.24 | |
| | Retired | 10.55±1.37 | | 8.50±0.94 | | 6.82±1.17 | |
| | Student | 9.35±1.45 | | 8.11±1.14 | | 6.22±1.25 | |
| | Unemployed | 9.54±1.65 | | 7.99±1.11 | | 6.33±1.18 | |

***Significant at 0.01 level

This observation is positive as females are more responsible for maintaining family hygiene and teaching their children. The retirees' highest attitude score (8.50±0.94) might be a reflection of their highest knowledge score. However, for professional groups, their knowledge score mismatched their attitude score. Surprisingly, the attitude scores of government staff (7.94 ±1.31) were the lowest. Homemakers had a better attitude score (8.42±0.76) than the unemployed, governments staff, professionals, and students.

The participants' mean practice score was poor (6.19±1.24 on a scale of 8.00) across all of the sociodemographic groups (Table 2). There was a significant difference between males and females (p<0.01) and occupational groups (p<0.01) with respect to the practice scores. Females (6.44±1.16), homemakers (6.55±1.12), and retirees (6.82±1.17) had higher practice scores than the other participants. Overall, statistically significant and positive linear correlations were observed between knowledge and attitude (r = 0.249, p<0.01) and attitude and practice (r = 0.148, p<0.01) (S4 Table).

## Assessment of KAP responses

The participants' frequency distribution on the KAP questions (Tables 3–5) demonstrated that 54.8% had factual knowledge of COVID-19 and identified it as a deadly disease, curable, and with a low mortality rate, which revealed that almost half were poorly informed about the disease, and 36.2% identified COVID-19 as a deadly disease with the certainty of death. A staggering 82.8% of the participants did not understand the cause of the COVID-19's emergence. Similarly, nCOV's contagiousness was unclear to one-fifth of the participants. Approximately 50% of the subjects believed that wearing surgical masks was effective for preventing infection, while 25% thought that masks were inadequate, and the rest demonstrated confusion. The participants demonstrated sound knowledge on COVID-19's symptoms (~99%), the need for every person to adopt preventive measures (~90%), the quarantine duration (95%), the methods of reducing the spread of COVID-19 (98%), and understanding the treatment (~98%), demonstrating the positive outcomes of awareness campaigns. The concept of quarantine was satisfactory in 86% of the participants while one-tenth wrongly considered that staying at home with family members equates quarantine. However, the participants' opinions varied markedly on the meaning of quarantine. The participants' knowledge on the risk of the spread of nCOV in Bangladesh compared to other countries was alarming, as the majority (75%) chose the wrong options while almost all (99%) were wrong in selecting the priority measures that the government needs to adopt to stop the spread. They also demonstrated a poor understanding of factors associated with the spread of nCOV as almost half (47%) chose the wrong options.

Table 4 shows the frequency distribution of the participants' attitudes toward COVID-19. As the majority (96%) were anxious about widespread COVID-19 fatalities in Bangladesh, most (95%) were willing to stay at home for two weeks upon government order. The attitude toward social distancing showed mixed outcomes as ~24% of the participants doubted its efficacy. The participants had a positive attitude toward stopping business and recreational trips (96.6%) and working from home (98.4%). Their attitudes regarding the government initiatives demonstrated pessimism, as 91.5% felt that adopting preventive measures was inadequate and 86% believed that public officials lagged in pre-emptive preparations after learning about the spread of the novel coronavirus from Wuhan.

The majority (~88%) of the participants thought that measures to protect healthcare professionals were inadequate as they (~88%) understood the elevated risks of COVID-19 among healthcare professionals. However, almost one-third of the participants (32%) were skeptical that the novel coronavirus would be widespread in Bangladesh. Overall, in stark contrast to

**Table 3. Frequency distribution of the responses to knowledge questions.**

| Questions | Responses | Frequency | Percentage |
|---|---|---|---|
| Perceptions about COVID-19 | | | |
| | A curse from the God | 122 | 7.68 |
| | A deadly disease with certainty of death | 575 | 36.19 |
| | A deadly disease, curable, with a low mortality rate | 870 | 54.75 |
| | A rumor that is being spread through public or media | 22 | 1.38 |
| COVID-19 emerged due to following reasons. Multiple answers are allowed | | | |
| | Right answers | 273 | 17.18 |
| | Wrong answers | 1316 | 82.82 |
| The main clinical symptoms of COVID-19 are fever, fatigue, dry cough, and difficulty breathing | | | |
| | Maybe | 176 | 11.08 |
| | No | 4 | 0.25 |
| | Yes | 1409 | 88.67 |
| Currently there is no effective cure for COVID-2019, but early diagnoses and supportive treatment can help most patients recover | | | |
| | Maybe | 293 | 18.44 |
| | No | 32 | 2.01 |
| | Yes | 1264 | 79.55 |
| Only seniors with chronic illnesses and other health complications are more likely to be seriously affected. | | | |
| | Maybe | 349 | 21.96 |
| | No | 425 | 26.75 |
| | Yes | 815 | 51.29 |
| People with COVID-2019 with no fever cannot infect others | | | |
| | Maybe | 171 | 10.76 |
| | No | 1262 | 79.42 |
| | Yes | 156 | 9.82 |
| COVID-19 spreads via respiratory droplets (from coughing and sneezing) of infected people | | | |
| | Maybe | 82 | 5.16 |
| | No | 25 | 1.57 |
| | Yes | 1482 | 93.27 |
| The general public can wear routine medical masks to prevent COVID-19 infection | | | |
| | Maybe | 397 | 24.98 |
| | No | 392 | 24.67 |
| | Yes | 800 | 50.35 |
| It is unnecessary for children and young adults to take measures to prevent COVID-19 | | | |
| | Maybe | 31 | 1.95 |
| | No | 1425 | 89.68 |
| | Yes | 133 | 8.37 |
| What do you understand about quarantine? | | | |
| | No clear understanding | 19 | 1.20 |
| | Stay in a separate room and have no contact with family members | 1368 | 86.09 |
| | Stay at home but can go outside | 23 | 1.45 |
| | Stay at home with family members | 180 | 11.33 |
| When should we quarantine? Multiple answers are allowed | | | |
| | Right answer | 102 | 6.42 |
| | Wrong answer | 1487 | 93.58 |
| Quarantine period | | | |
| | Right answer | 1516 | 95.41 |
| | Wrong answer | 73 | 4.59 |

(*Continued*)

**Table 3.** (Continued)

| Questions | Responses | Frequency | Percentage |
|---|---|---|---|
| Isolating and treating COVID-19 patients are effective methods of reducing the spread of the virus | | | |
| | Maybe | 130 | 8.18 |
| | No | 28 | 1.76 |
| | Yes | 1431 | 90.06 |
| Compared with other affected nations, what is the possibility of COVID-19 spreading in Bangladesh? (Multiple answers) | | | |
| | Right answers | 399 | 25.11 |
| | Wrong answers | 1190 | 74.89 |
| What could be the possible reasons for COVID-19's spread in Bangladesh if it happens? (Multiple answers) | | | |
| | Right answers | 842 | 52.99 |
| | Wrong answers | 747 | 47.01 |
| What should be the government's priorities to control the spread of COVID-19? (Multiple answers) | | | |
| | Right answers | 9 | 0.57 |
| | Wrong answers | 1580 | 99.43 |

their dissenting attitude toward the public sectors' readiness related to the adequacy of preventive measures (91.5%), need for pre-emptive measures (~80%), or safeguarding healthcare providers (~88%), the participants showed a positive attitude toward healthcare providers in perceiving their risks and the need to protect them (~88%).

On the practice side (Table 5), as the majority of the participants (~92%) were somewhat stressed about COVID-19 in Bangladesh, leading to a high to extreme feeling about the risk (65%), 80.5% reported that they avoided crowded areas and 94% did not allow their children to engage in outdoor activities and preferred wearing masks (~84%) when going out. Overall, ~60% of the participants were working from home full-time while ~14% occasionally worked from home. This may not reflect reality in Bangladesh since the Internet-based survey excluded responses from lower-income people for whom working from home is not an option. Approximately 61% of the participants thought that the public awareness level was low or increasing, which may explain their mixed opinions regarding COVID-19-related panic in their respective areas.

Analysis of attitudes and practices in relation to knowledge and sociodemographic variables.

As shown in Table 6, 34.8% of the participants belonged to the "poor knowledge with poor attitude" group and 37.1% to the "poor knowledge with poor practice" group. Only 33% of the participants demonstrated good knowledge while 52.4% had good attitudes and 44.8% showed good practices. Among the participants with good knowledge, 38.7% exhibited poor attitudes and 55.2% demonstrated poor practices.

The logistic regression analysis also showed that the participants with a better understanding of COVID-19 favored social distancing (odds ratio [OR] 1.65; 95% confidence interval (CI):1.26–2.15; p<0.01) or working from home (OR 1.71; 95% CI:1.20–2.44; p<0.01) (Table 7). They had a better attitude toward the seriousness of the threat to healthcare providers (OR 1.83; 95% CI:1.28–2.63; p<0.01) but demonstrated a general dissatisfaction toward the government's early response to COVID-19 (OR 1.46; 95% CI:1.10–1.90; p<0.01) and provision of protection for healthcare workers (OR 1.46; 95% CI:1.04–2.06; p<0.05).

Women were more willing than men to maintain social distance (OR 1.45; 95% CI:1.12–1.87; p<0.01), cancel trips (OR 1.42; 95% CI:1.11–1.82; p<0.01), and work from home (OR 2.10; 95% CI:1.49–2.97; p<0.01). They also expected better pre-emptive responses from the government (OR 2.00; 95% CI:1.52–2.63; p<0.01) and perceived a disproportionate threat of COVID-19 to healthcare workers (OR 1.56; 95% CI:1.12–2.18; p<0.01). Among the age

**Table 4. Frequency distribution of the responses to attitude questions.**

| Questions | Responses | Frequency | Percentage |
|---|---|---|---|
| Will you stay at home for a certain period (14 days) to prevent the spread of COVID-19 if government orders? | | | |
| | Yes | 1512 | 95.15 |
| | No | 18 | 1.13 |
| | Not possible due to work | 59 | 3.71 |
| Do you think that social distancing (for example, staying 1–2 m apart, avoiding crowds, etc.) can prevent the spread of COVID-19? | | | |
| | Maybe | 300 | 18.88 |
| | No | 74 | 4.66 |
| | Yes | 1215 | 76.46 |
| Do you agree that we should cancel business/recreational trips at this time? | | | |
| | Maybe | 33 | 2.08 |
| | No | 21 | 1.32 |
| | Yes | 1535 | 96.60 |
| Do you believe that working from home can help control COVID-19? | | | |
| | Maybe | 162 | 10.20 |
| | No | 25 | 1.57 |
| | Yes | 1402 | 88.23 |
| Do you agree that the government has taken sufficient preventive measures to prevent the spread of COVID-19? | | | |
| | No | 823 | 51.79 |
| | Not enough | 631 | 39.71 |
| | Yes | 135 | 8.50 |
| Do you agree that the government should have taken preventive measures when COVID-19 was first reported in China? | | | |
| | Maybe | 92 | 5.79 |
| | No | 222 | 13.97 |
| | Yes | 1275 | 80.24 |
| Do you think that COVID-19 can cause widespread fatalities in Bangladesh? | | | |
| | Maybe | 295 | 18.57 |
| | No | 59 | 3.71 |
| | Yes | 1235 | 77.72 |
| Do you believe that COVID-19 will not be an epidemic in Bangladesh due to following reasons? (Multiple answers) | | | |
| | Right answers | 1081 | 68.03 |
| | Wrong answers | 508 | 31.97 |
| Do you agree that our healthcare providers (for example, doctors, nurses, and support staff) are under serious threat when they treat infected people? | | | |
| | Maybe | 102 | 6.42 |
| | No | 89 | 5.60 |
| | Yes | 1398 | 87.98 |
| Do you think that the government has ensured enough protective measures for healthcare providers? | | | |
| | Maybe | 119 | 7.49 |
| | No | 1396 | 87.85 |
| | Yes | 74 | 4.66 |

groups, in reference to the 18–25 age group, remaining three groups 26–35 (OR 1.39; 95% CI:1.06–1.81; p<0.05), 36–45 (OR 1.64; 95% CI:1.09–2.45; p<0.05), and older than 45 (OR 2.20; 95% CI:1.34–3.63; p<0.01) were willing to cancel business/recreational trips due to COVID-19. As expected, in reference to government staff, professionals showed a higher reluctance toward the government's measures (OR 2.92; 95% CI:1.48–5.77; p<0.01). Interestingly, students (OR 0.38; 95% CI:0.18–0.81; p<0.05) and unemployed (OR 0.38; 95% CI:0.17–0.85; p<0.05) participants were less willing to engage in social distancing.

**Table 5. Frequency distribution of the responses to practices questions.**

| Questions | Responses | Frequency | Percentage |
|---|---|---|---|
| Do you presently go to crowded areas? | | | |
| | Every day | 14 | 0.88 |
| | No | 1279 | 80.49 |
| | Sometimes | 268 | 16.87 |
| | Yes | 28 | 1.76 |
| Do you allow your children to engage in outdoor activities? | | | |
| | No | 1498 | 94.27 |
| | Sometimes | 78 | 4.91 |
| | Yes | 13 | 0.82 |
| Do you and family members use masks when outside? | | | |
| | No | 90 | 5.66 |
| | Sometimes | 240 | 15.10 |
| | Yes | 1259 | 79.23 |
| Have you started working from home in the last few weeks due the COVID-19 outbreak? | | | |
| | No | 420 | 26.43 |
| | Sometimes | 229 | 14.41 |
| | Yes | 940 | 59.16 |
| How would you rate the awareness level of those living around you regarding COVID-19? | | | |
| | Awareness level is increasing | 265 | 16.68 |
| | Low awareness but increasing | 724 | 45.56 |
| | No precautionary measures undertaken at all | 176 | 11.07 |
| | People around me are highly aware and careful | 70 | 4.41 |
| | Some precautionary measures have been taken | 354 | 22.28 |
| How would you rate the medical facilities in Bangladesh to manage COVID-19? | | | |
| | Gradual advancement in healthcare is noticeable to manage COVID-19 | 77 | 4.85 |
| | Health facilities are available for a limited number of people | 381 | 23.98 |
| | Medical facilities are highly appropriate and can prevent the spread of COVID-19 | 11 | 0.69 |
| | Bangladesh has good facilities to prevent COVID-19 | 9 | 0.57 |
| | Very poor facilities are available thus far | 1111 | 69.92 |
| Are those in your area/district already panicking about COVID-19? | | | |
| | Maybe | 338 | 21.27 |
| | No | 302 | 19.01 |
| | Yes | 949 | 59.70 |
| Are you feeling anxious/stressed/depressed/helpless thinking about the COVID-19 outbreak? | | | |
| | Maybe | 117 | 7.36 |
| | No | 129 | 8.12 |
| | Yes | 1343 | 84.52 |
| If above answer is YES, then please rate your level of feelings. | | | |
| | Extreme | 276 | 20.55 |
| | High | 612 | 45.57 |
| | Little | 66 | 4.91 |
| | Moderate | 389 | 28.97 |

As evident from the logistic regression analysis, the participants' knowledge was reflected in some of the practices. The participants with good knowledge did not allow their children to engage in outdoor activities during COVID-19 (OR 1.75; 95% CI:1.06–2.89; p<0.05) (Table 8). The female participants were more concerned than the males about visiting crowded

**Table 6. Cross tabulation of good and poor attitudes and practices with respect to the participants' COVID-19 knowledge status.**

| | | | Knowledge | |
|---|---|---|---|---|
| | | | Good (%) | Poor (%) |
| Attitudes | Good (n) | n (column%) (% of total) | 321 (61.3) (20.2) | 512(48.1) (32.2) |
| | Poor (n) | n (column%) (% of total) | 203 (38.7) (12.8) | 553 (51.9) (34.8) |
| Practices | Good (n) | n (column%) (% of total) | 235 (44.8) (14.8) | 476 (44.7) (30.0) |
| | Poor (n) | n (column%) (% of total) | 289 (55.2) (18.2) | 589 (55.3) (37.1) |

areas (OR 2.96; 95% CI:2.16–4.05; p<0.01), allowing their children to engage in outdoor activities (OR 2.06; 95% CI:1.27–3.34; p<0.01), and wearing a face mask when going outside (OR 1.31; 95% CI:1.00–1.71; p<0.05). They were also more anxious than males (OR 2.19; 95%

**Table 7. Multiple logistic regressions of different variables of attitudes with socio-demographic variable and knowledge status.** (Values are odds ratio followed by 95% confidence interval in parenthesis).

| Variables in for multiple logistic regression | | Social distancing | Cancel business / recreational trips | Working from home | Sufficient preventive measures by Govt. | Response from Govt. after reports from Wuhan | Massive fatality or not | Seriousness of threat to healthcare providers | Protection for healthcare providers |
|---|---|---|---|---|---|---|---|---|---|
| Gender | | | | | | | | | |
| | Male | 1.00 | 1.00 | 1.00 | | 1.00 | 1.00 | 1.00 | 1.00 |
| | Female | 1.45 (1.12–1.87)*** | 1.42 (1.11–1.82)*** | 2.10 (1.49–2.97)*** | | 2.00 (1.52–2.63)*** | 0.77 (0.58–1.02) | 1.56 (1.12–2.18)*** | 0.91 (0.67–1.23) |
| Age (years) | | | | | | | | | |
| | 18–25 | 1.00 | 1.00 | | 1.00 | | 1.00 | 1.00 | |
| | 26–35 | 0.97 (0.67–1.41) | 1.39 (1.06–1.81)** | | 0.72 (0.40–1.28) | | 1.29 (0.84–1.97) | 1.27 (0.89–1.83) | |
| | 36–45 | 0.96 (0.57–1.65) | 1.64 (1.09–2.45)** | | 0.35 (0.16–0.76)*** | | 1.72 (1.00–2.98) | 0.90 (0.56–1.47) | |
| | Over 45 | 1.20 (0.65–2.24) | 2.20 (1.34–3.63)*** | | 0.62 (0.25–1.57) | | 1.77 (0.98–3.22) | 0.75 (0.44–1.27) | |
| Occupation | | | | | | | | | |
| | Govt. staff | 1.00 | | | 1.00 | | 1.00 | | |
| | Home makers | 0.75 (0.24–2.29) | | | 2.56 (0.68–9.64) | | 0.66 (0.22–2.02) | | |
| | Professionals | 0.60 (0.30–1.21) | | | 2.92 (1.48–5.77)*** | | 1.12 (0.61–2.04) | | |
| | Retired | $^\lambda$ 1.21 (0.13–10.97) | | | $^\lambda$ 3.12E8 (0.00–3.1E8) | | 0.71 (0.13–3.76) | | |
| | Student | 0.38 (0.18–0.81)** | | | 1.22 (0.53–2.80) | | 1.19 (0.59–2.40) | | |
| | Unemployed | 0.38 (0.17–0.85)** | | | 1.85 (0.72–4.79) | | 0.75 (0.33–1.68) | | |
| Knowledge | | | | | | | | | |
| | Poor | 1.00 | | 1.00 | | 1.00 | | 1.00 | 1.00 |
| | Good | 1.65 (1.26–2.15)*** | | 1.71 (1.20–2.44)*** | | 1.45 (1.10–1.90)*** | | 1.83 (1.28–2.63)*** | 1.46 (1.04–2.06)** |
| Constant | | 5.09*** | 2.29*** | 5.02*** | 8.187*** | 2.85*** | 0.19*** | 5.07*** | 6.72*** |

***Significant at 0.01 level

**Significant at 0.05 level (2-tailed); $^\lambda$ Higher OR values were due to small sample size and similar responses; Blank cells reveal that these variables were excluded from logistic regression analysis because these were not significant in chi-squared test.

**Table 8. Multiple logistic regressions of different variables of practices with socio-demographic variable and knowledge status.** (Values are odds ratio followed by 95% confidence interval in parenthesis).

| Variables in for multiple logistic regression | | Presently do not visit crowded areas | Do not allow children to engage in outdoor activities | Do not wear a face mask when going outside | Level of awareness among the neighbors | Rating the medical facilities | Panic among neighbors | Anxious/ stressed/ due to COVID-19 | Stress level/ anxiety due to COVID-19 |
|---|---|---|---|---|---|---|---|---|---|
| Gender | | | | | | | | | |
| | Male | 1.00 | 1.00 | 1.00 | 1.00 | | 1.00 | 1.00 | 1.00 |
| | Female | 2.96 (2.16–4.05)*** | 2.06 (1.27–3.34)*** | 1.31 (1.00–1.71)** | 0.82 (0.66–1.02) | | 0.76 (0.59–0.99)** | 2.19 (1.60–3.00)*** | 1.01 (0.82–1.25) |
| Age (years) | | | | | | | | | |
| | 18–25 | 1.00 | | 1.00 | 1.00 | 1.00 | 1.00 | 1.00 | |
| | 26–35 | 0.91 (0.60–1.37) | | 1.28 (0.86–1.90) | 1.12 (0.80–1.56) | 1.15 (0.89–1.49) | 1.01 (0.68–1.50) | 1.33 (0.96–1.83) | |
| | 36–45 | 1.14 (0.66–1.98) | | 1.06 (0.62–1.81) | 1.09 (0.70–1.71) | 1.79 (1.19–2.70)*** | 1.20 (0.68–2.12) | 1.62 (0.99–2.65) | |
| | Over 45 | 1.14 (0.63–2.07) | | 0.847 (0.48–1.50) | 1.13 (0.69–1.85) | 1.37 (0.88–2.13) | 2.41 (1.13–5.12) | 0.71 (0.46–1.12) | |
| Occupation | | | | | | | | | |
| | Govt. staff | 1.00 | | 1.00 | 1.00 | | 1.00 | | 1.00 |
| | Home makers | 2.87 (0.78–10.58) | | 0.53 (0.21–1.36) | 1.01 (0.46–2.23) | | 1.04 (0.35–3.05) | | 1.36 (0.60–3.08) |
| | Professionals | 1.52 (0.90–2.57) | | 0.90 (0.49–1.67) | 1.00 (0.60–1.64) | | 0.84 (0.42–1.69) | | 1.85 (1.09–3.12)** |
| | Retired | λ5.11 (0.60–43.83) | | λ4.9E8 (0.00–4.9E8). | 0.50 (0.13–1.85) | | 0.86 (0.09–8.00) | | λ2.78 (0.77–10.09) |
| | Student | 2.15 (1.15–4.05)** | | 0.87 (0.43–1.75) | 1.20 (0.68–2.13) | | 0.73 (0.34–1.57) | | 1.35 (0.80–2.29) |
| | Unemployed | 3.05 (1.43–6.48)*** | | 0.90 (0.42–1.94) | 1.67 (0.88–3.16) | | 0.71 (0.31–1.61) | | 1.91 (1.02–3.56)** |
| Knowledge | | | | | | | | | |
| | Poor | | 1.00 | | | 1.00 | | | |
| | Good | | 1.75 (1.06–2.89)** | | | 0.83 (0.65–1.05) | | | |
| Constant | | 1.62 | 11.17*** | 3.63*** | 1.51 | 2.76*** | 5.66*** | 3.78*** | 0.43*** |

***Significant at 0.01 level

**Significant at 0.05 level (2-tailed); λ Higher OR values were due to small sample size and similar responses; Blank cells reveal that these variables were excluded from logistic regression analysis because these were not significant in chi-squared test.

CI:1.60–3.00; p<0.01) and perceived fewer signs of panic among their neighbors (OR 0.76; 95% CI:0.59–0.99; p<0.05). The participants in the 36–45 age group were more aware of existing medical facilities for treating COVID-19 patients compared to the 18–25 age group (OR 1.79; 95% CI:1.19–2.70; p<0.01). In reference to government staff, students (OR 2.15; 95% CI:1.15–4.05; p<0.05) and unemployed (OR 3.05; 95% CI:1.43–6.48; p<0.01) participants were more willing to avoid crowded areas, and both professionals (OR 1.85; 95% CI:1.09–3.12; p<0.05) and the unemployed (OR 1.91; 95% CI:1.02–3.56; p<0.05) had a higher level of anxiety due to COVID-19.

## Discussion

In this section, we discuss the overall KAP scores followed by responses to KAP questions and the relationships among attitudes, practices, and knowledge.

## Knowledge, attitudes, and practices' scores

Overall, this study's participants demonstrated poor knowledge scores toward COVID-19. Only 33% had good knowledge of COVID-19 whereas the older subjects, specifically retirees, were more knowledgeable on COVID-19 than the younger participants, particularly students who had a lower knowledge score. Studies in countries with similar sociodemographics reported higher knowledge scores. For example, in Nepal, 84.25% of respondents demonstrated good knowledge [39] whereas in India, Iran, and Pakistan this was 80.64%, 85%, and 64.8%, respectively [40–42]. However, research in Malaysia and Saudi Arabia reported that seniors had higher knowledge than younger people, similar to our study [43,44]. There might be several reasons for poor COVID-19 knowledge among this study's participants. There were only a few COVID-19 cases in Bangladesh when this study was conducted, so many were unfamiliar with this disease, and awareness campaigns also likely fell short in reaching all groups equally.

The average attitude score indicated the desired attitude toward COVID-19 among the participants, and females had a better attitude than males. Similar results were also reported in Saudi Arabia [43], India [41], and Sudan [45]. Surprisingly, the attitude scores of government staff and students were the lowest rather than homemakers.

In comparison to attitude, the mean practice score was poor across all of the sociodemographic groups, inconsistent with COVID-19 related KAP studies elsewhere [40–41,46]. The low practice score in this study strongly indicates the gap in the translation of knowledge and attitudes into practices. Poor knowledge, attitudes, and practices among the students and public service professionals are concerning since young people constitute a substantial percentage of the population while public service professionals execute public policies and mitigation responses. Hence, special preference should be given to increase these two groups' knowledge while focusing on improving their attitudes and practices. The observed differences may be due to disproportionate exposure of different groups to media and information-gathering networks, among other factors. For example, while students and young adults are more engaged in social media, the seniors and homemakers have more time to spend watching television and gather better information to convert it into better practices. This indicates avenues of improvement in an awareness campaign to target appropriate age groups and professional groups through appropriate media.

## Responses to KAP questions

Regarding COVID-19 and its causes, many of the participants (36.2%) considered it a deadly disease with a certainty of death and had wrong perceptions about the cause of the emergence (82.8%) of COVID-19. These observations indicate the effect of misinformation from the Internet and media on their understanding of the cause of the emergence of COVID-19. Also, disagreement among the participants regarding the risk of seniors with comorbid diseases is indicative of an inadequate understanding of the outcomes of COVID-19. Many of the participants were confused about wearing surgical masks, which is linked to policymakers' indecision and mixed messages regarding the use of masks. These results were commensurate with observations from instances of harsh treatment of those who are either diagnosed with COVID-19 or showing symptoms [36,47]. The results demonstrate that disease symptoms, preventive measures, quarantine duration, treatment understanding were well answered by the participants, although there were some doubts about how and where to quarantine. But most of the participants (99.43%) disagreed about priority and preventive measures that the government has implemented to stop the spread of nCOV, especially preparations lacking for healthcare workers' safety (87.85%). These observations are explained by newspaper reports [24,48]

related to peoples' behavior contradicting the measures needed to control COVID-19 and overwhelming doubts on measures taken by the government. This indicates that it is necessary to better communicate information to educate the public so that they can better understand government policies and respond positively by conforming to the right attitudes and practices.

The results clearly indicated high to extremely high social anxiety (66.12%) due to the high fear dying of coronavirus but the participants were confused about staying home and maintaining social distancing. However, the actual situation differs. The government has deployed law enforcement to implement lockdown and social distancing [49]. The attitude scores and related responses clearly indicated a bleak perception toward the government's readiness and ability to manage COVID-19. Experts also opined that there was a lack of coordination toward the management of COVID-19 in Bangladesh although we had adequate time to implement appropriate measures [27]. As the majority of the participants anticipated a high possibility of the spread of COVID-19 with a high fatality rate, the situation likely created moderate to extreme stress or anxiety among most of the subjects. Policy makers need to take heed of this to address the psychological aspects of the pandemic.

## Relations among attitudes, practices, and knowledge

The practice of any population regarding a particular issue depends on their understanding of the issue and attitude toward it. Interestingly, among the participants with poor knowledge scores, one-third had good practice scores. These results indicate the challenges of managing the pandemic on the part of the government both in policy making and implementing mitigation measures since even those with good knowledge may not behave accordingly. This is also reflected in participants' expectations of high fatality and contagion while showing poor awareness of the reasons for contagion. Unfortunately, most of the participants could not understand the priorities of the government's actions to prevent COVID-19 while dissenting on its preparedness. The government of Bangladesh implemented necessary measures and ordered at-home quarantines of expatriates returning from COVID-19 infected countries [24,50–51]. However, violations and even protests against the government's orders were evident in many cases [37], with the excuse of inappropriate institutional quarantine facilities [52]. Gender, age, occupation, and knowledge scores had a strong effect on the participants' opinions on both attitude and practice responses in logistic regression analysis. A similar observation was reported in Sudan and Peru [45,53]. This clearly indicates that policy makers must better educate the public through awareness campaigns as knowledge creates a more positive attitude to harness better practice and encourage constructive criticism.

## Limitations of this study

This study is one of the first in Bangladesh assessing peoples' knowledge, attitudes, and practices toward COVID-19 and thus provides a useful baseline for future research. Further research can address some of its methodological limitations. One of this study's key limitations is that the number of participants from whom the samples were collected is unknown. As a result, we could not estimate the response rate or determine the sample size prior to data collection. We used an English version of the questionnaire and so those did not understand English could not participate in the online survey. A Bengali (national language) questionnaire would increase the responses and represent a wider sociodemographic distribution. This study mainly targeted participants with a university-level education and so the findings cannot represent the KAP of Bangladeshi general public. This is one of the major limitations of online research [40,43,44,53]. A wider study comprising participants from wider sociodemographics helps overcome this limitation.

## Conclusion

Knowledge, attitudes, and practices of the general public are crucial to contain COVID-19 and implement mitigation measures. An understanding of the public's KAP helps policy makers and public health managers design and implement policies and mitigation measures by providing them with insights into pertinent crucial factors. This study demonstrated poor COVID-19 KAP among the participants, with some significant effects of the sociodemographic factors on the scores. This study indicated a need for more curated awareness programs with differential targeting and messages for varying sociodemographic groups, especially students and public service professionals. As 99% of the participants failed to identify the government's priorities to combat the disease, policy makers must communicate more transparently to improve public confidence regarding factual information on preventive measures and their effectiveness so that people do not panic and spontaneously follow measures. This study suggests that routine KAP analysis can be an effective monitoring tool to measure the performance of mitigation measures in COVID-19. In any such application, the results of this study can be used as a baseline in Bangladesh.

## Supporting information

**S1 Table. KAP questionnaire for the COVID-19 online survey in Bangladesh.**
(DOC)

**S2 Table. Cut point of knowledge, attitudes, and practices.**
(DOCX)

**S3 Table. Categories of responses to attitudes and practices used for logistic regression.**
(DOC)

**S4 Table. Pearson's correlation coefficients for knowledge, attitudes, and practices.**
(DOCX)

**S1 Data. KAP-COVID-19-Raw Data.**
(XLSX)

## Acknowledgments

We thank the anonymous participants for volunteering to participate in this study. We also thank Dr. Kannan Navaneetham, academic editor (PLOS ONE), and the reviewers for their constructive criticism and comments on our manuscript.

## Author Contributions

**Conceptualization:** Alak Paul, Dwaipayan Sikdar, Tapan Kumar Nath.

**Data curation:** Mohammad Mosharraf Hossain, Janardan Mahanta.

**Formal analysis:** Mohammad Mosharraf Hossain, Janardan Mahanta.

**Investigation:** Alak Paul, Dwaipayan Sikdar, Md Robed Amin, Farah Deeba, Tapan Kumar Nath.

**Methodology:** Janardan Mahanta, Tapan Kumar Nath.

**Project administration:** Alak Paul, Tapan Kumar Nath.

**Supervision:** Alak Paul, Tapan Kumar Nath.

**Writing – original draft:** Dwaipayan Sikdar, Mohammad Mosharraf Hossain, Farah Deeba, Md. Akib Jabed, Mohammad Mohaiminul Islam, Sharifa Jahan Noon, Tapan Kumar Nath.

**Writing – review & editing:** Alak Paul, Dwaipayan Sikdar, Mohammad Mosharraf Hossain, Md Robed Amin, Farah Deeba, Tapan Kumar Nath.

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
