## [Decision Letter · Decision Letter 0]

22 Jun 2020

PONE-D-20-12138

Knowledge, attitude and practice towards the novel corona virus among Bangladeshi people: Implications for mitigation measures

PLOS ONE

Dear Dr. Nath,

Thank you for submitting your manuscript to PLOS ONE. After careful consideration, we feel that it has merit but does not fully meet PLOS ONE’s publication criteria as it currently stands. Therefore, we invite you to submit a revised version of the manuscript that addresses the points raised during the review process.

We look forward to receiving your revised manuscript.

Kind regards,

Kannan Navaneetham, PhD

Academic Editor

PLOS ONE

Journal Requirements:

"The study has been financed by the team of researchers involved in the work, and no external funding was available."

"The author(s) recieved no specific funding for this work"

3. We note that Figure 'S1 Fig' in your submission contain map images which may be copyrighted. All PLOS content is published under the Creative Commons Attribution License (CC BY 4.0), which means that the manuscript, images, and Supporting Information files will be freely available online, and any third party is permitted to access, download, copy, distribute, and use these materials in any way, even commercially, with proper attribution. For these reasons, we cannot publish previously copyrighted maps or satellite images created using proprietary data, such as Google software (Google Maps, Street View, and Earth). For more information, see our copyright guidelines: http://journals.plos.org/plosone/s/licenses-and-copyright.

3.1.    You may seek permission from the original copyright holder of Figure 'S1 Fig' to publish the content specifically under the CC BY 4.0 license.

Reviewers' comments:

Reviewer's Responses to Questions

**Comments to the Author**

1. Is the manuscript technically sound, and do the data support the conclusions?

Reviewer #1: Yes

Reviewer #2: Partly

2. Has the statistical analysis been performed appropriately and rigorously? 

Reviewer #1: Yes

Reviewer #2: No

3. Have the authors made all data underlying the findings in their manuscript fully available?

Reviewer #1: Yes

Reviewer #2: No

4. Is the manuscript presented in an intelligible fashion and written in standard English?

Reviewer #1: No

Reviewer #2: No

5. Review Comments to the Author

Reviewer #1: 1. The authors should go throughout the manuscript and carefully scrutinize many grammatical errors and punctuations. The authors should ask a native speaker of English to improve the manuscript and abstract.

2. This manuscript reports knowledge, attitude and practice toward COVID-19 among general population of Bangladesh. Under COVID-19 outbreak all over the world, the topic is really important. However, design of this study and structure have some issues and limitations. The authors should separate the results from the discussion section for better understanding of the manuscript.

3. Although COVID-19 outbreak is a recent global issue, the authors should support the discussion of their study by recent articles regarding KAP towards COVID-19 in other areas. articles concerning KAP towards COVID-19 have been recently published both in neighboring countries and other parts of the world. The authors should revise their discussion section and add more comparison to similar studies in other areas. (e.g: Erfani A, Knowledge, Attitude and Practice toward the Novel Coronavirus (COVID-19) Outbreak: A Population-Based Survey in Iran.2020 / Zhong B-L et al Knowledge, attitudes, and practices towards COVID-19 among Chinese residents during the rapid rise period of the COVID-19 outbreak: a quick online cross-sectional survey. Int J Biol Sci. 2020 / Srichan P, et al. Knowledge, Attitude and Preparedness to Respond to the 2019 Novel Coronavirus (COVID-19) Among the Bordered Population of Northern Thailand in the Early Period of the Outbreak: A Cross-Sectional Study. 2020)

4. The authors should revise the conclusion to a shorter and clearer version. as much of the information in this section are more appropriate for the discussion section.

5. The authors should add a section regarding their limitation in their study, e.g. the selection bios which particularly consists of higher educated individuals, etc.

6. One of the big drawbacks of this study is the method of sample collection. The authors used a web-based survey; however, the details of recruitment were unclear. To call participants, the authors seemed to provide the questionnaire via email and Facebook, but if so, who saw the link? What is the response rate? In other words, what is the “mother population” of this survey? This study had a serious selection bias problem along with small sample size. The aim of this study was to understand the status of knowledge regarding COVID-19. The sampling method in this study is not suitable for this purpose under the unclear mother population.

7. The authors allocated the same scoring for incorrect and “Maybe”. What is the reason and validity of these allocations? Since the Likert scale scoring system differentiates “don’t know” from “incorrect” answer. In other words, does incorrect knowledge and no knowledge have the same impact on disease control and transmission among the population? If the authors allocated different scoring for “incorrect answers” and “don’t know” would there be any difference in results?

8. The authors failed to provide a framework for the analyzing section of the data. The work requires an extremely detailed use of language throughout the paper to ensure that shared meaning is held among the readers and the authors. The authors tended to rely on vague terms and need more stringent attention to detail in language to better support the authors’ suppositions.

9. The authors need to explain in the method section how the sample size was determined prior to data gathering. In other words, what was the estimated sample size?

10. The authors mention in their ethical statement that the participants provided informed consents. Please clarify how these consents where obtained (oral or written).

Reviewer #2: This paper addressed the KAP about COVID-19 among Bangladeshi people. The authors conducted a cross-sectional study and received complete response from 1589 participants. Considering socioeconomic and literacy in Bangladesh, the findings of this study is very significant. However, I have some major concerns.

- This study may not be representative of Bangladeshi population as only 4.2% of the participants were recruited from the cohort with education level secondary or below and 95% from university graduate. However, the prevalence of university graduate in the country is may be less than 10%. Thus the findings of this study may not represent Bangladeshi population, instead it may represent Bangladeshi people who have completed university degree. Since, English is not a mother language, the English version of the questionnaire further increases selection/participation bias in the study.

- I would recommend to report the actual distribution of education as well as occupation level in Bangladesh,

- The reported data in line 253 do not match with the data provided in the Table 2.

- The multiple logistic regression analysis results in Table 7A and 7B are confusing. Clearly, the analysis has not been done for attitude or practice score; instead it has been done for various component of attitude and practice. This should be reflected from the title of the table. Some of the outcome variables in these table have more than two categories. I am wondering how the logistic regression analysis was performed for these outcomes.

- In the above table, I notice some reported ORs are exceptionally very high (e.g., Table 7A, OR related to Age>65yrs is 3.211E7). How do you interpret this OR. This arises because there was not enough participants in this age group. The authors may combine this age group with the preceding group.

- I would recommend to report 95% CI for reported OR in Table 7A & B.

- I am concern that the authors choose to present results and discussion in the same section.

- I find the conclusion section very long. I would recommend to summarise the findings in a short section.

- What are the limitation of this study; this need to be stated.

6. PLOS authors have the option to publish the peer review history of their article (what does this mean?). If published, this will include your full peer review and any attached files.

Reviewer #1: Yes: Reza Shahriarirad

Reviewer #2: No

---

## [Author Response · Author response to Decision Letter 0]

22 Jul 2020

Response to Reviewers

PONE-D-20-12138

Knowledge, attitude and practice towards the novel corona virus among Bangladeshi people: Implications for mitigation measures

We thank the academic editor and the reviewers for their feedback. We found all the reviewers’ comments to be usefully constructive, and we appreciate the time taken to provide such thoughtful and thorough feedback. We would also like to thank the reviewers for their positive comments. Both reviewers, however, made some suggestions around ways to improve the manuscript which we found useful and have addressed.

In the following sections, we provide more detail on these changes, addressing the reviewers’ comments one by one. 

Editor’s Comments

Comments 1: A rebuttal letter that responds to each point raised by the academic editor and reviewer(s). You should upload this letter as a separate file labeled 'Response to Reviewers'.

Response: We addressed all comments raised by the editor and reviewers and uploaded a file labeled “Response to Reviewers”.

Comments 2: A marked-up copy of your manuscript that highlights changes made to the original version. You should upload this as a separate file labeled 'Revised Manuscript with Track Changes'.

Response: We have done so accordingly.

Comments 3: An unmarked version of your revised paper without tracked changes. You should upload this as a separate file labeled 'Manuscript'.

Response: We have done so accordingly.

Comments 4: Response: We have not made any changes to financial disclosure.

Comments 5: Guidelines for resubmitting your figure files are available below the reviewer comments at the end of this letter.

Response: We followed the guidelines

Comments 6: If applicable, we recommend that you deposit your laboratory protocols in protocols.io to enhance the reproducibility of your results. Protocols.io assigns your protocol its own identifier (DOI) so that it can be cited independently in the future. For instructions see: http://journals.plos.org/plosone/s/submission-guidelines#loc-laboratory-protocols

Response: Our study does not have any laboratory protocol to deposit.

Comments 7: When submitting your revision, please ensure that your manuscript meets PLOS ONE's style requirements, including those for file naming. The PLOS ONE style templates can be found at

 Response: We followed PLOS ONE style in this revision.

Comments 8: Thank you for stating the following in the Funding Section of your manuscript:

"The study has been financed by the team of researchers involved in the work, and no external funding was available."

"The author(s) received no specific funding for this work"

 Response: We removed funding statement from the revised manuscript

Comments 9: We note that Figure 'S1 Fig' in your submission contain map images which may be copyrighted. All PLOS content is published under the Creative Commons Attribution License (CC BY 4.0), which means that the manuscript, images, and Supporting Information files will be freely available online, and any third party is permitted to access, download, copy, distribute, and use these materials in any way, even commercially, with proper attribution. For these reasons, we cannot publish previously copyrighted maps or satellite images created using proprietary data, such as Google software (Google Maps, Street View, and Earth). For more information, see our copyright guidelines: http://journals.plos.org/plosone/s/licenses-and-copyright.

Response: We removed “Figure S1” from revised manuscript

Reviewers' comments:

Reviewer #1: 

Comments 1: The authors should go throughout the manuscript and carefully scrutinize many grammatical errors and punctuations. The authors should ask a native speaker of English to improve the manuscript and abstract.

Response: This revised paper was edited by Elsevier English Editing Service (a certificate is attached)

Comments 2: This manuscript reports knowledge, attitude and practice toward COVID-19 among general population of Bangladesh. Under COVID-19 outbreak all over the world, the topic is really important. However, design of this study and structure have some issues and limitations. The authors should separate the results from the discussion section for better understanding of the manuscript.

Response: We understand that submitted manuscript has some issues to be improved. We tried our best to improve this paper based on reviewers’ comments, added a section on Limitations of this Study. We also separated Results and Discussion sections.

Comments 3: Although COVID-19 outbreak is a recent global issue, the authors should support the discussion of their study by recent articles regarding KAP towards COVID-19 in other areas. articles concerning KAP towards COVID-19 have been recently published both in neighboring countries and other parts of the world. The authors should revise their discussion section and add more comparison to similar studies in other areas. (e.g: Erfani A, Knowledge, Attitude and Practice toward the Novel Coronavirus (COVID-19) Outbreak: A Population-Based Survey in Iran.2020 / Zhong B-L et al Knowledge, attitudes, and practices towards COVID-19 among Chinese residents during the rapid rise period of the COVID-19 outbreak: a quick online cross-sectional survey. Int J Biol Sci. 2020 / Srichan P, et al. Knowledge, Attitude and Preparedness to Respond to the 2019 Novel Coronavirus (COVID-19) Among the Bordered Population of Northern Thailand in the Early Period of the Outbreak: A Cross-Sectional Study. 2020).

Response: Thanks a lot for helping us with important references. We discussed our results with above and additional references.

Comment 4: The authors should revise the conclusion to a shorter and clearer version. as much of the information in this section are more appropriate for the discussion section.

Response: We revised Conclusions in a shorter form.

Comments 5: The authors should add a section regarding their limitation in their study, e.g. the selection bios which particularly consists of higher educated individuals, etc.

Response: We have added a section on Limitations of this Study where we described all of our shortcomings of the study.

Comments 6: One of the big drawbacks of this study is the method of sample collection. The authors used a web-based survey; however, the details of recruitment were unclear. To call participants, the authors seemed to provide the questionnaire via email and Facebook, but if so, who saw the link? What is the response rate? In other words, what is the “mother population” of this survey? This study had a serious selection bias problem along with small sample size. The aim of this study was to understand the status of knowledge regarding COVID-19. The sampling method in this study is not suitable for this purpose under the unclear mother population.

Response: We do agree that sampling was one of the limitations of this study. Mother population was unknown. The survey link was circulated to Facebook friends and emailed to people known to authors of this paper. As population was unknown, it was not possible to estimate the response rate. We addressed this in Methods section.

Comments 7: The authors allocated the same scoring for incorrect and “Maybe”. What is the reason and validity of these allocations? Since the Likert scale scoring system differentiates “don’t know” from “incorrect” answer. In other words, does incorrect knowledge and no knowledge have the same impact on disease control and transmission among the population? If the authors allocated different scoring for “incorrect answers” and “don’t know” would there be any difference in results?

Response: In this revision, we changed the scoring approach. We used score 1 for each correct answer, 0.5 for maybe answer and 0 for each incorrect answer. With this new scoring approach we revised Table 2.

Comments 8: The authors failed to provide a framework for the analyzing section of the data. The work requires an extremely detailed use of language throughout the paper to ensure that shared meaning is held among the readers and the authors. The authors tended to rely on vague terms and need more stringent attention to detail in language to better support the authors’ suppositions.

Response: Our apology. We do not understand the comment “The authors failed to provide a framework for the analyzing section of the data”. However, we elaborated Methods and Data Analysis. As stated above, the revised paper was checked by a professional English editor.

Comments 9: The authors need to explain in the method section how the sample size was determined prior to data gathering. In other words, what was the estimated sample size?

Response: As our “mother population” was unknown, we could not determine sample size prior to data gathering. In fact, we stopped receiving responses when there were very low survey participation.

Comments 10: The authors mention in their ethical statement that the participants provided informed consents. Please clarify how these consents where obtained (oral or written).

Response: We stated following statement to clarify this: “Before participating in the survey, respondents had to answer a Yes/No question to confirm their consent to participate voluntarily. After conformation of this question, the respondents were directed to the complete the questionnaire”. 

Reviewer #2

Comments 1: This paper addressed the KAP about COVID-19 among Bangladeshi people. The authors conducted a cross-sectional study and received complete response from 1589 participants. Considering socioeconomic and literacy in Bangladesh, the findings of this study is very significant. However, I have some major concerns.

Response: Thanks for your kind notes.

Comments 2: This study may not be representative of Bangladeshi population as only 4.2% of the participants were recruited from the cohort with education level secondary or below and 95% from university graduate. However, the prevalence of university graduate in the country is may be less than 10%. Thus the findings of this study may not represent Bangladeshi population, instead it may represent Bangladeshi people who have completed university degree. 

Response: We completely agree with this comment. As the respondents were authors’ Facebook friends and friends of friends’, they were mostly educated. This was one of the limitations of this study, and researcher addressed this as one of the limitations of online study.

Comments 3: Since, English is not a mother language, the English version of the questionnaire further increases selection/participation bias in the study.

Response: This was true, and one of our study limitations.

Comments 4: I would recommend to report the actual distribution of education as well as occupation level in Bangladesh,

Response: From Bangladesh Statistical data, we have current distribution of education and occupation level in Bangladesh as follow:

Education: Primary (67.89%), Secondary (30.95%) and post-secondary (1.16%). In case of occupation, the distribution is Agriculture (38.58%), Industry (21.26%) and Service sector (40.16%) (Source: Bangladesh Bureau of Statistics, 2017)

Comments 5: The reported data in line 253 do not match with the data provided in the Table 2.

Response: We have corrected this.

Comments 6: The multiple logistic regression analysis results in Table 7A and 7B are confusing. Clearly, the analysis has not been done for attitude or practice score; instead it has been done for various component of attitude and practice. This should be reflected from the title of the table. Some of the outcome variables in these table have more than two categories. I am wondering how the logistic regression analysis was performed for these outcomes.

Response: Absolutely right. The analysis was not done for attitudes/practices’ scores, but for various attributes of attitudes and practices. We have revised the Table caption to avoid the confusion.

For Logistic regression, variable were categorized into two categories. All “Yes” responses as YES, and all “No and Maybe” responses as OTHERS for a number of questions. These YES and OTHERS categories were used for regression analysis.

Comments 7: In the above table, I notice some reported ORs are exceptionally very high (e.g., Table 7A, OR related to Age>65yrs is 3.211E7). How do you interpret this OR. This arises because there was not enough participants in this age group. The authors may combine this age group with the preceding group.

Response: This is absolutely valid comment. In this revision, we have combined the suggested age groups.

Comments 8: I would recommend to report 95% CI for reported OR in Table 7A & B.

Response: We have provided 95% CI in Table 7A & 7B.

Comments 9: I am concern that the authors choose to present results and discussion in the same section.

Response: We have separated Results and Discussion

Comments 10: I find the conclusion section very long. I would recommend to summarise the findings in a short section.

Response: We have shorten Conclusion 

Comments 11: What are the limitation of this study; this need to be stated.

Response: We have added a section on Limitations of this Study

---

## [Decision Letter · Decision Letter 1]

4 Aug 2020

PONE-D-20-12138R1

Knowledge, attitudes and practices toward the novel coronavirus among Bangladeshis: Implications for mitigation measures

PLOS ONE

Dear Dr. Nath,

Thank you for submitting your manuscript to PLOS ONE. After careful consideration, we feel that it has merit but does not fully meet PLOS ONE’s publication criteria as it currently stands. Therefore, we invite you to submit a revised version of the manuscript that addresses the points raised during the review process.

We look forward to receiving your revised manuscript.

Kind regards,

Kannan Navaneetham, PhD

Academic Editor

PLOS ONE

Reviewers' comments:

Reviewer's Responses to Questions

**Comments to the Author**

1. If the authors have adequately addressed your comments raised in a previous round of review and you feel that this manuscript is now acceptable for publication, you may indicate that here to bypass the “Comments to the Author” section, enter your conflict of interest statement in the “Confidential to Editor” section, and submit your "Accept" recommendation.

Reviewer #1: All comments have been addressed

Reviewer #2: All comments have been addressed

2. Is the manuscript technically sound, and do the data support the conclusions?

Reviewer #1: Yes

Reviewer #2: Partly

3. Has the statistical analysis been performed appropriately and rigorously? 

Reviewer #1: Yes

Reviewer #2: Yes

4. Have the authors made all data underlying the findings in their manuscript fully available?

Reviewer #1: Yes

Reviewer #2: Yes

5. Is the manuscript presented in an intelligible fashion and written in standard English?

Reviewer #1: Yes

Reviewer #2: Yes

6. Review Comments to the Author

Reviewer #1: The authors have done a great job in addressing the concerns and comments and have significantly improved the manuscript. However, a few concerns still remain which can help improve the manuscript before considering it for publication.

1. In “table 2” and “supplementary table 4” please report <0.001 instead of 0.000

2. I recommend adjusting the design of the tables, particularly the borders, since there is no consistency and seems that the borders are inserted randomly. An option can be using three-line tables.

3. In table 7 what do the numbers represent? They seem to be OR and 95%CI, if so, please state in the table.

4. The conclusion section of the manuscript still needs improvement and can be shortened and improved, as it should demonstrate the final result and what the authors have achieved and how it can be used or implanted.

5. I recommend reporting the “cumulative percentage” in the first table of the “supplementary 2” file as it is more valuable to the readers. The authors could even express the table as a cumulative frequency/percentage graph if they choose.

Reviewer #2: (No Response)

7. PLOS authors have the option to publish the peer review history of their article (what does this mean?). If published, this will include your full peer review and any attached files.

Reviewer #1: **Yes: **Reza Shahriarirad

Reviewer #2: **Yes: **Md Billah

---

## [Author Response · Author response to Decision Letter 1]

7 Aug 2020

Response to Reviewers

PONE-D-20-12138R1

Knowledge, attitudes and practices toward the novel coronavirus among Bangladeshis: Implications for mitigation measures

We thank the academic editor and the reviewers for their further feedback. We found all the reviewers’ comments to be usefully constructive, and we appreciate the time taken to provide such thoughtful and thorough feedback. Reviewers made some more suggestions to improve the manuscript which we found useful and have addressed.

In the following sections, we provide more detail on these changes, addressing the reviewers’ comments one by one. 

Editor’s Comments

Comments: Thank you for submitting your manuscript to PLOS ONE. After careful consideration, we feel that it has merit but does not fully meet PLOS ONE’s publication criteria as it currently stands. Therefore, we invite you to submit a revised version of the manuscript that addresses the points raised during the review process.

Response: Thanks for considering our revised paper for further revision. We have addressed all comments of reviewers.

Comments: A rebuttal letter that responds to each point raised by the academic editor and reviewer(s). You should upload this letter as a separate file labeled 'Response to Reviewers'.

Response: We have addressed all comments and uploaded a file named “Response to Reviewers”.

Comments: A marked-up copy of your manuscript that highlights changes made to the original version. You should upload this as a separate file labeled 'Revised Manuscript with Track Changes'.

Response: We have uploaded a marked-up copy named “Revised Manuscript with Track Changes”.

Comments: An unmarked version of your revised paper without tracked changes. You should upload this as a separate file labeled 'Manuscript'.

Response: We have uploaded revised paper named “Manuscript”.

Reviewers' comments

Reviewer #1: 

Comments: The authors have done a great job in addressing the concerns and comments and have significantly improved the manuscript. However, a few concerns still remain which can help improve the manuscript before considering it for publication.

Response: Thanks for your kind comments. We have addressed all concerns in this revision (R2).

Comments: In “table 2” and “supplementary table 4” please report <0.001 instead of 0.000.

Response: We have corrected accordingly.

Comments: I recommend adjusting the design of the tables, particularly the borders, since there is no consistency and seems that the borders are inserted randomly. An option can be using three-line tables.

Response: We re-designed all Tables following your comments.

Comments: In table 7 what do the numbers represent? They seem to be OR and 95%CI, if so, please state in the table.

Response: The numbers represent odds ratio (OR) followed by 95% CI in parenthesis.

Comments: The conclusion section of the manuscript still needs improvement and can be shortened and improved, as it should demonstrate the final result and what the authors have achieved and how it can be used or implanted.

Response: We have tried to shortened conclusion.

Comments: I recommend reporting the “cumulative percentage” in the first table of the “supplementary 2” file as it is more valuable to the readers. The authors could even express the table as a cumulative frequency/percentage graph if they choose.

Response: We provided cumulative percentage in Table S2.

---

## [Editor Report · Decision Letter 2]

12 Aug 2020

PONE-D-20-12138R2

Knowledge, attitudes and practices toward the novel coronavirus among Bangladeshis: Implications for mitigation measures

PLOS ONE

Dear Dr. Nath,

Thank you for submitting your manuscript to PLOS ONE. After careful consideration, we feel that it has merit but does not fully meet PLOS ONE’s publication criteria as it currently stands. Therefore, we invite you to submit a revised version of the manuscript that addresses the points raised during the review process.

We look forward to receiving your revised manuscript.

Kind regards,

Kannan Navaneetham, PhD

Academic Editor

PLOS ONE

Additional Editor Comments (if provided):

Table 2: footnote- should be *** Significant at 0.01 level; Remove Less than or equal to. Also change in other tables 7A and 7B also.

Pages 17-21: 95%CI is missing in the interpretation of results for all covariates. CI should be included. For example, Page 17, 3rd line of the last paragraph: "working from home (OR 1.708, 95% CI, p<0.01)"- This should be written as "working from home (OR=1.708, 95%CI:1.198-2.437, p<0.01). Include CI in all places. Follow the PLOS ONE guidelines or refer previously published articles in PLOS ONE.

Table 7A and 7B: Include in the Table Title at the end within a bracket- Values are Adjusted Odds Ratios followed by 95% Confidence Interval in parenthesis, instead of in the footnote.

Table 7A and 7B: For all reference category, include OR is 1.00 (NO CI) instead of leaving it blank.

Table 7B and 7B still needs proper formatting. All are mixed. For example, Table 7A: Gender and Age are mixed. Check other places also. They are not formatted properly. I would also appreciate, if Odd ratios are given at two decimal places both in the table and in the text.

Table 7A: Var-sufficient preventive measures by govt. Odd ratios for Retired category is too high. Could be due small sample issues. Check it and give foot note on the inappropriate result.

Table 7B: same problem- Odd ratio too high for retired for outcome variable- wear a face mask when going outside.

Table 7A and &B: Several Cells are missing. Give a footnote why there is no OR for those variables.

Table 7B: Clearly define the outcome variables. The interpretations and the definition of outcomes are not consistent. For example, page 20, para 2, "The participants with good knowledge did not allow their children to

engage in outdoor activities during COVID-19 (OR 1.751, 95% CI, p<0.05) (Table 7B). According to the table 7B column heading (Presently visit crowded areas, if this is coded as 1), having good knowledge is greater odds to visit crowded areas than those with poor knowledge (reference category). Similarly check other interpretations on page 20 and 21.

---

## [Author Response · Author response to Decision Letter 2]

16 Aug 2020

Response to Reviewers

PONE-D-20-12138R2

Knowledge, attitudes and practices toward the novel coronavirus among Bangladeshis: Implications for mitigation measures

We thank the academic editor for further feedback. We found all comments to be usefully constructive, and we appreciate the time taken to provide such thoughtful and thorough feedback. We have addressed all suggestions to improve the manuscript.

In the following sections, we provide more detail on these changes, addressing the comments one by one. 

Editor’s Comments

Comments: Thank you for submitting your manuscript to PLOS ONE. After careful consideration, we feel that it has merit but does not fully meet PLOS ONE’s publication criteria as it currently stands. Therefore, we invite you to submit a revised version of the manuscript that addresses the points raised during the review process.

Response: Thanks for considering our revised paper for further revision. We have addressed all comments in this revision.

Comments: A rebuttal letter that responds to each point raised by the academic editor and reviewer(s). You should upload this letter as a separate file labeled 'Response to Reviewers'.

Response: We have addressed all comments and uploaded a file named “Response to Reviewers”.

Comments: A marked-up copy of your manuscript that highlights changes made to the original version. You should upload this as a separate file labeled 'Revised Manuscript with Track Changes'.

Response: We have uploaded a marked-up copy named “Revised Manuscript with Track Changes”.

Comments: An unmarked version of your revised paper without tracked changes. You should upload this as a separate file labeled 'Manuscript'.

Response: We have uploaded revised paper named “Manuscript”.

Comments: Table 2: footnote- should be *** Significant at 0.01 level; Remove Less than or equal to. Also change in other tables 7A and 7B also.

Response: We have changed accordingly in Table 2, Table 7, and Table 8

Comments: Pages 17-21: 95%CI is missing in the interpretation of results for all covariates. CI should be included. For example, Page 17, 3rd line of the last paragraph: "working from home (OR 1.708, 95% CI, p<0.01)"- This should be written as "working from home (OR=1.708, 95%CI:1.198-2.437, p<0.01). Include CI in all places. Follow the PLOS ONE guidelines or refer previously published articles in PLOS ONE.

Response: We have added values of 95% Confidence Interval in results following a recent PLOS ONE paper.

Comments: Table 7A and 7B: Include in the Table Title at the end within a bracket- Values are Adjusted Odds Ratios followed by 95% Confidence Interval in parenthesis, instead of in the footnote.

Response: We have split Table 7A and 7B into Table 7 and Table 8, and moved ‘values are adjusted odds ratios followed by 95% Confidence Interval in parenthesis at the end of Table title. 

Comments: Table 7A and 7B: For all reference category, include OR is 1.00 (NO CI) instead of leaving it blank.

Response: We provided OR 1.00 accordingly.

Comments: Table 7B and 7B still needs proper formatting. All are mixed. For example, Table 7A: Gender and Age are mixed. Check other places also. They are not formatted properly. I would also appreciate, if Odd ratios are given at two decimal places both in the table and in the text.

Response: We have re-formatted Table 7 & 8, and odd ratios are given up to two decimal places.

Comments: Table 7A: Var-sufficient preventive measures by govt. Odd ratios for Retired category is too high. Could be due small sample issues. Check it and give foot note on the inappropriate result.

Table 7B: same problem- Odd ratio too high for retired for outcome variable- wear a face mask when going outside.

Response: We provided a reason for these higher OR values “higher OR values were due to small sample size and their responses were similar). 

Comments: Table 7A and &B: Several Cells are missing. Give a footnote why there is no OR for those variables.

Response: We provided following foot note “Blank cells reveal that these variables were excluded from logistic regression analysis because these were not significant in chi-squared test.”.

Comments: Table 7B: Clearly define the outcome variables. The interpretations and the definition of outcomes are not consistent. For example, page 20, para 2, "The participants with good knowledge did not allow their children to engage in outdoor activities during COVID-19 (OR 1.751, 95% CI, p<0.05) (Table 7B). According to the table 7B column heading (Presently visit crowded areas, if this is coded as 1), having good knowledge is greater odds to visit crowded areas than those with poor knowledge (reference category). Similarly check other interpretations on page 20 and 21.

Response: We have re-worded the outcomes variables in Table 8. Now the interpretations are consistent with outcomes variables.

---

## [Editor Report · Decision Letter 3]

19 Aug 2020

Knowledge, attitudes and practices toward the novel coronavirus among Bangladeshis: Implications for mitigation measures

PONE-D-20-12138R3

Dear Dr. Nath,

We’re pleased to inform you that your manuscript has been judged scientifically suitable for publication and will be formally accepted for publication once it meets all outstanding technical requirements.

Kind regards,

Kannan Navaneetham, PhD

Academic Editor

PLOS ONE
---

## [Editor Report · Acceptance letter]

24 Aug 2020

PONE-D-20-12138R3 

Knowledge, attitudes, and practices toward the novel coronavirus among Bangladeshis: Implications for mitigation measures 

Dear Dr. Nath:

I'm pleased to inform you that your manuscript has been deemed suitable for publication in PLOS ONE. Congratulations! Your manuscript is now with our production department. 

Kind regards, 

on behalf of

Professor Kannan Navaneetham 

Academic Editor

PLOS ONE